# Binding Free Energy Analysis of Colicin D, E3 and E8 to Their Respective Cognate Immunity Proteins Using Computational Simulations

**DOI:** 10.3390/molecules30061277

**Published:** 2025-03-12

**Authors:** Mahesh Koirala, Clifton K. Fagerquist

**Affiliations:** 1Department of Agriculture, Produce Safety & Microbiology, Western Regional Research Center, Agricultural Research Service, U.S., Albany, CA 94710, USA; clifton.fagerquist@usda.gov; 2Department of Energy, Research Participation Program Administered by the Oak Ridge Institute for Science and Education, U.S., Oak Ridge, TN 37830, USA

**Keywords:** colicins, immunity proteins, molecular dynamics simulations, MM-PBSA, protein–protein interactions, binding free energy, AlphaFold2, food safety

## Abstract

Colicins are antimicrobial proteins produced by bacteria for the purpose of destroying neighboring bacteria. Colicin activity is neutralized by a specific cognate immunity protein in order to protect the host. This study investigates the structural and binding mechanisms underlying the interaction of colicin-D, -E3 and -E8 to their respective immunity proteins (ImD, Im3 and Im8) using structure prediction, molecular dynamics (MD) simulations and MM-PBSA approach of free energy calculations. High-confidence colicin-immunity (Col-Im) complex structures predicted using AlphaFold2 were subjected to MD simulations of 150 ns with GROMACS and were analyzed for the binding free energy calculation using gmx_MMPBSA. Results showed that the complex of Col_E3-Im3 exhibited the most favorable binding free energy, driven by strong van der Waals and electrostatic interactions. Col_D-ImD and Col_E8-Im8 also showed the favorable binding. Electrostatics and hydrogen bonding emerged as a key factor driving binding and stability, while polar solvation acted as a destabilizing factor across all systems. These outcomes provide an understanding of the molecular mechanisms of Col-Im systems, with potential applications for developing natural antimicrobials for food safety.

## 1. Introduction

Colicins are plasmid-encoded proteins secreted by pathogenic bacteria to attack competing bacteria that occupy the same local environment [1]. Colicins destroy the DNA, RNA (t-RNA or ribosomal RNA) or outer membrane of neighboring bacteria. For colicin to not attack the bacterial host that produced it, an immunity gene is co-expressed with the colicin gene. The immunity protein strongly binds to its cognate colicin until it is secreted from the host cell. Upon attachment of the colicin/immunity complex to the outer membrane receptor of a neighboring bacterial cell, the complex undergoes rearrangement such that the immunity protein detaches from its colicin partner. Although there is an associated metabolic burden to the bacterial host for plasmid carriage, plasmid genes can provide a survival benefit to the host, especially in environments where bacteria compete for limited resources. Colicins effectively act as natural antibiotics deployed by bacteria against bacterial competitors [1].

The only defense against colicin is the cognate immunity protein, whose plasmid-encoded genes must be expressed by the bacterium under attack. A plasmid-encoded immunity gene may be expressed in excess (or large excess) beyond that of the 1:1 stoichiometric ratio of colicin-to-immunity in order to have sufficient immunity protein in the event of an external colicin attack, i.e., a basal expression of the immunity gene [2]. However, colicin and their cognate immunity genes are under SOS regulatory control, which means their genes are repressed by a DNA-binding protein (LexA) that binds to a DNA inverted repeat (SOS box), blocking the expression of genes downstream [3]. The SOS response is triggered by DNA damage: single-stranded DNA, UV radiation or DNA-damaging antibiotics. RecA is the master regulator of the SOS response, and its initiation results in the self-cleavage of LexA, thus unblocking the expression of colicin and immunity genes [3]. It is essential that a colicin nuclease, once translated, is inactivated so that it does not destroy host DNA or RNA. In consequence, the immunity protein must recognize and bind quickly and tightly to its cognate colicin. While the colicin and immunity genes are co-expressed, it is essential to have, at least, a slight excess of the immunity protein so that the colicin is quickly inactivated while inside the host cell.

Colicin-bearing plasmids (D, E3 and E8) were found to be quite prevalent amongst environmental isolates of pathogenic Shiga toxin-producing *E. coli* (STEC) collected from agricultural regions and previously analyzed by our group [4,5,6,7]. As such, there is a need to better understand these three distinct colicin nucleases, whose targets are tRNA, rRNA, and DNA, respectively. E3 and E8 attach to the same outer membrane receptor BtuB, whereas D gains entry by attachment to FepA [1]. In the current study, we have calculated the binding energies of immunity proteins: ImD, Im3 and Im8 to their respective colicins: D, E3 and E8 to better understand the interactions between colicins and their cognate immunity proteins.

### 1.1. Protein–Protein Interactions

Protein–protein interactions (PPIs) are essential to every biological process and act as the basis for signal transduction, cellular organization, immune responses, and metabolic regulation. Proteins interact, forming the complexes associated in a precise manner, which are critical for the proper functioning of the cells and organisms. Several diseases, including cancer, neurogenerative disorders, and infection, occur due to irregularities in these interactions, making PPIs a major aspect of biomedical research [8,9,10,11,12]. The structural complexity of PPIs represents challenges as well as opportunities in drug discovery. Many PPIs have historically been classified as “undruggable” due to the large, flat binding surfaces that lack clear pockets for small-molecule drugs. However, recent advances in high-throughput screening, computational modeling, and fragment-based drug discovery have enabled scientists to target these previously subtle interactions [10]. Moreover, PPIs are increasingly recognized as potential therapeutic targets in diseases where protein interactions drive pathological processes, such as the p53-MDM2 interaction in cancer [13,14]. PPIs also play a crucial role in microbial competition, which is of great interest in the area of food production and safety. Bacterial proteins like colicins obstruct the growth of competing microorganisms and influence microbial ecology. Colicins and their corresponding immunity proteins are known to contribute to bacterial fitness, leading to harnessing and improving food safety by targeting harmful pathogens in foodborne bacteria [1]. Understanding these interactions is essential for developing novel biocontrol approaches to boost food safety and avoid adulteration in agricultural systems.

Recent developments in computational techniques and resources, including molecular dynamics (MD) simulations, machine learning (ML) algorithms, and protein structure prediction using artificial intelligence (AI), have modernized and advanced the study of PPIs [15]. Tools like AlphaFold have dramatically improved the accuracy of protein structure prediction, while MD simulations allow for the investigation of PPI dynamics and binding free energy calculations in unprecedented detail [16]. These computational tools, alongside experimental approaches like X-ray crystallography, cryo-electron microscopy, nuclear magnetic resonance (NMR), and native state mass spectrometry, provide a comprehensive view of PPI networks and their role in cellular function [17,18,19,20].

### 1.2. Binding Free Energy

Binding free energy is a critical thermodynamic quantity that provides insights into the stability and affinity of molecular complexes, including protein–protein, protein–ligand and protein–DNA/RNA interactions. It is crucial for understanding molecular recognition, biological interactions, and complex formation, which are crucial in drug design, protein engineering, and biomolecular studies. Molecular dynamics (MD) simulations combined with free energy calculation methods have proven to be computationally effective tools for estimating these energies [21,22,23].

The binding free energy (∆*G_bind_*) is represented as the free energy difference between the bound complex and the unbound components. It can be stated mathematically as:
(1)$$∆G_{bind}=G_{complex}-G_{prot1}-G_{prot2}$$
where $G_{complex}$ is the free energy of the bound complex, $G_{prot1}$ and $G_{prot2}$ are the free energies of the unbound proteins separately. Various computational approaches exist to estimate binding free energy. Among these, Molecular Mechanics Poisson-Boltzmann Surface Area (MM-PBSA) and Molecular Mechanics Generalized Born Surface Area (MM-GBSA) methods are widely used due to their balance of computational efficiency and accuracy.

### 1.3. Molecular Mechanics Poisson-Boltzmann Surface Area (MM-PBSA)

In this method, the binding free energy is approximated by summing the gas-phase molecular mechanics (MM) energy, solvation-free energies, and an entropic term. The binding free energy is calculated as:
(2)$$∆G_{bind}={∆E}_{MM}+{∆G}_{solv}-T∆S$$

In which
(3)$$∆E_{\mathrm{MM}}=\Delta E_{int}+{∆E}_{elec}+{∆E}_{vdw}$$
(4)$$∆G_{solv}={∆G}_{polar}+∆G_{non-polar}$$
where ∆E_MM_ is the difference in molecular mechanics energy (van der Waals and electrostatic interactions). ∆*G_solv_* is the solvation-free energy through polar contributions estimated using the Poisson-Boltzmann (PB) equation and nonpolar contributions valued based on surface area (SA). *T*∆*S* denotes the entropic involvement, typically estimated using normal mode analysis, quasi-harmonic analysis, or Schlitter’s formula [24].

### 1.4. Molecular Mechanics Generalized Born Surface Area (MM-GBSA)

This method is like MM-PBSA but uses the Generalized Born (GB) model to evaluate solvation-free energy instead of the PB equation. MM-GBSA follows the same general formula as MM-PBSA, with the solvation term (∆*G_solv_*) computed using the GB model. These approaches use the representative conformational snapshots of the molecular system obtained from molecular dynamics simulations. With the average of the generated snapshots, MM-PBSA and MM-GBSA provide approximate estimates of the binding free energy by integrating the dynamic behavior of the molecules [21,25]. Additionally, the entropy term (−*T*∆*S*) is essential for estimating the overall binding free energy. Accurate entropy calculations can be computationally challenging and more complex. Different approaches, such as normal mode analysis, Schlitter’s entropy [24], and quasi-harmonic approximation, are frequently used to estimate this term [26].

In the current study, we focus on the interaction between bacterial protein complexes, specifically colicin D, E3 and E8 and their immunity protein cognates. These protein complexes are crucial to microbial competition, bacterial survival and regulation within bacterial communities, directly affecting the safety of food production environments. We use molecular dynamics simulations and free energy calculations, providing insights into the structural and energetic mechanisms guiding the stability and specificity of these protein complexes. The findings may provide a significant contribution to a better understanding of PPIs and open new paths for food safety research.

## 2. Results

### 2.1. Structure of Colicin/Immunity Complexes

The structural details of the three colicin/immunity protein complexes colicin D-immunity D (Col_D-ImD), colicin E3-immunity E3 (Col_E3-Im3), and colicin E8-immunity E8 (Col_E8-Im8) predicted from AlphaFold2 (Figure 1a, Figure 2a and Figure 3a) reveal both similarities and unique features that contribute to their binding mechanisms and the neutralization of colicin toxicity. It is found that all the colicins (Col_D, Col_E3, and Col_E8) have extended helical structures that play a vital role in defining their architecture. The smaller immunity proteins (ImD, Im3, and Im8) bind tightly to critical regions of the colicins, temporarily disabling their toxic effects. This binding is a key structural feature that ensures the immunity proteins can effectively inhibit the enzymatic function (ribonuclease or DNase) of the colicins, allowing the host to be protected. The electrostatic interactions between the colicins and their cognate immunity protein are shown (Figure 1b, Figure 2b and Figure 3b), with negatively charged regions shown in red color and positively charged regions shown in blue color. In Col_D-ImD, ImD has distinct positively and negatively charged regions that lie within the charge complementary of Col_D. This kind of electrostatic interaction is essential for guiding the tight binding between the two proteins, maintaining the stability of the complex. Similarly, in Col_E3-Im3, Im3 shows tight electrostatic interaction with Col_E3 near the ribonuclease domain. The charge distribution on Im3 is critical for stabilizing the binding interaction that temporarily disables Col_E3 toxicity. A closer view of the electrostatic potential highlights the charge distribution at the binding interface of Col_E8-Im8, where Im8 wraps around the DNase domain of Col_E8. This electrostatic mapping between two regions of negative and positive charges further stabilizes their interaction. Hence, all three complexes separately show electrostatic complementarity, playing a central role in driving the binding interactions.

Another important feature of these interactions is the number of hydrogen bonds between the colicins and their corresponding immunity proteins (Table 1). Col_D-ImD represents the robust intermolecular interactions with 13 initial hydrogen bonds. Where Col_E3-Im3 presents 11 hydrogen bonds in the native state, Col_E8-Im8 forms only 10 hydrogen bonds. This reflects the relatively weaker interaction in Col_E8-Im8 compared to Col_E3-Im3. This helps to underscore the importance of hydrogen bonding in determining the binding strength of the colicin/immunity complexes. The differences in charge distribution and bond formation confirm that each immunity protein can only bind to and neutralize its respective colicin, providing a clear example of how electrostatic potential drives protein–protein interactions in these systems. Understanding these structural and electrostatic features provides valuable insight into the protective mechanisms that immunity proteins use to safeguard against colicin toxicity.

### 2.2. MD Simulations

MD simulations provide comprehensive insights into how a protein interacts with its biological partner, offering a holistic understanding crucial for guiding protein–protein interactions. The root-mean-square deviation (RMSD) analysis of the colicin/immunity complexes over the last 50 ns of the simulation shows robust stability and convergence for all complexes, as shown in Figure 4. Col_D-ImD shows the structural stability and consistent equilibrium dynamics with RMSD varying between 0.8 and 1.2 nm across both independent runs. With RMSD values ranging from 0.6 to 1.2 nm, Col_E3-Im3 shows moderate fluctuations, highlighting the stable dynamics and comparable trends. Meanwhile, in Col_E8-Im8, RMSD values vary between 0.6 and 1.0 nm, relating slightly more pronounced variations but suggesting stability with dynamic movement throughout the simulation period. These results demonstrate that all three complexes maintain overall structural stability during the analyzed timeframe. The RMSD cluster distribution analysis for Colicin D, E3, and E8 complexes (Figure 5) provides insights into their conformational stability over 100–150 ns of the simulations. The clustering analysis revealed that each complex primarily occupies a few dominant conformational states, with the largest cluster consistently capturing most of the trajectory frames. Colicin D-ImD exhibited a well-converged distribution with minimal variation between runs, whereas Colicin E3-Im3 showed noticeable differences between Run 1 and Run 2, indicating a broader sampling of conformations. In contrast, Colicin E8-Im8 displayed a more compact cluster distribution, highlighting stable interaction interfaces, confirming that the simulations adequately captured the relevant conformational space and supporting the robustness of the chosen simulation time.

The hydrogen bond (H-bond) analysis and average number of buried water molecules provide a deeper insight into the intermolecular interactions within the colicin-immunity complexes, as depicted in Table 1 and Figure 6. In Col_D-ImD, the average number of hydrogen bonds increases from 13 before the simulation to 15.83 during the last 50 ns, with notable fluctuations between 15 and 25 and indicates stable interaction with some flexibility. The average number of 20.15 buried water molecules reveals that interfacial stabilization depends on water-mediated hydrogen bonds to compensate for its flexibility. In Col_E3-Im3, the average number of hydrogen bonds rises from 11 initially to 13.56, with variation between 10 and 22. This reflects a more compact interface as supported by the average 23.55 buried water molecules that balance hydration and direct interactions. Col_E8-Im8 shows a decrease in average hydrogen bonds from 10 to 8.5 during the same timeframe, ranging between 10 and 18 and exhibited the lowest hydration with just 11.81 average water molecules. This reflects a tightly packed and stable interface dominated by direct protein–protein interactions. Thus, a clear map of the relation between hydrogen bonds, hydration and interfacial compactness is shown, highlighting reduced hydration correlating with tighter packing as seen in Col_E8-Im8. On the other hand, moderate hydration contributes to stability and flexibility, as seen in Col_D-ImD and Col_E3-Im3. The interfacial hydrogen bonds are also visualized and compared (Figure 7) for the last frame of each complex. This structural representation illustrates hydrogen bond strength with labeled distance, providing insight into relative strength and stability. Stronger interactions are indicated by shorter distances (1.7–2.2 Å), while weaker bonds appear at greater distances (>2.5 Å). The difference in the interaction landscape across the complexes also highlights the different interfacial stabilization mechanisms.

The root-mean-square fluctuation (RMSF) analysis for the colicin/immunity complexes, provided in the Appendix A, demonstrates residue-level flexibility for both the colicin and immunity proteins across two independent simulation runs. In Col_D-ImD, the Col_D protein displays higher fluctuations at the N- and C-terminal regions, along with some loop regions, while the ImD protein shows lower fluctuations with peaks around loop and interaction regions. Col_E3-Im3 also demonstrates significant flexibility in the Col_E3 protein around residue 300, indicating the dynamic loop regions, while the Im3 protein exhibits moderate flexibility with peaks at specific residues. In Col_E8-Im8, the Col_E8 protein shows pronounced variations around residue 300, whereas the Im8 protein maintains relatively low flexibility, with a peak observed at the C-terminal region.

Additionally, the solvent-accessible surface area (SASA) analyses for the complexes are also provided in Appendix A, which give insights into the compactness and exposure of the protein complexes over the last 50 ns of simulation. In Col_D-ImD, SASA displays fluctuations between 395 and 420 nm^2^, with consistent patterns across both runs, reflecting stable solvent exposure. With SASA values ranging from 285 to 315 nm^2^, Col_E3-Im3 shows a steady exposure of surface area across the runs. Also, Col_E8-Im8 maintains relatively constant SASA values between 310 and 340 nm^2^, highlighting consistent solvent interaction throughout the simulations.

The radius of gyration (Rg) in Appendix A shows Col_D-ImD has Rg values fluctuating around 4.6 to 5.2 nm, indicating stable compactness with slight variations between two runs. Col_E3-Im3 complex shows Rg values varying from 3.6 to 3.8 nm, with consistent patterns reflecting a stable structural arrangement. Similarly, the Col_E8-Im8 maintains Rg values between 3.5 and 3.9 nm, reflecting the sustained compactness and structural integrity over the simulation timeframe.

These results, consistent across the two independent simulation runs, provide additional validation of the structural stability and dynamics of the colicin/immunity complexes and are presented in the Appendix A for detailed reference.

### 2.3. Binding Free Energy from gmx_MMPBSA

The binding free energy analysis of the colicin/immunity complexes obtained from gmx_MMPBSA presents the significant contributions of various energy components to their stability and interactions. Two independent runs used from two trajectories of random seeds show convergence and reproducibility. Col_D-ImD shows the total binding free energy of −95.9 ± 0.20 kcal/mol (Run 1) and −77.2 ± 0.17 kcal/mol (Run 2), as shown in Figure 8. This favorable binding is predominantly driven by strong van der Waals and electrostatic interactions, which are essential for stabilizing the complex. However, the polar solvation energy is the destabilizing factor with partial counteraction by nonpolar solvation contributions. Similarly, Col_E3-Im3 resulted in the most favorable total binding free energy of −135.8 ± 0.19 kcal/mol (Run 1) and −144.7 ± 0.18 kcal/mol (Run 2), emphasizing the significant role of robust van der Waals and electrostatic interactions for stabilization. Despite the destabilizing effects of polar solvation energy, these interactions dominate and ensure a highly stable binding. While Col_E8-Im8 complex shows comparative lower total binding free energies of −54.4 ± 0.16 kcal/mol (Run 1) and −57.9 ± 0.17 kcal/mol (Run 2), still, van der Waals and electrostatic interactions drive the binding, with destabilization counteraction from polar solvation energy. Per residue decomposition of the binding free energy of interfacial residues are shown in the Appendix A highlighting the dominant contributions from van der Waals and electrostatic interactions.

The results shown provide fundamental insights into the binding mechanisms of colicin/immunity complexes, where van der Waals and electrostatic interactions emerge as the primary driving forces of the binding stability for each system, and polar solvation acts as a destabilizing factor but is mitigated by nonpolar solvation contributions. The most negative binding free energy of the Col_E3-Im3 underscores its robust intermolecular interactions, making it the most stable complex. These results help to highlight the roles of the molecular forces governing colicin-immunity binding, with potential implications for their biological roles and applications in food safety. The consistent trends observed across independent simulation runs further signify the reliability of the results and their broader applicability.

## 3. Discussion

We studied colicin-immunity protein interactions’ structural integrity and dynamics through molecular dynamics simulations and free energy calculations approaches. The consistent binding behaviors highlighting the driving forces provide insights into molecular stability and activity. The RMSD-based cluster analysis confirms the sample convergence into a few dominant clusters within the last 100–150 ns, supporting the sufficiency of the 150 ns simulation for the relative binding energy comparison. The Col_E3-Im3 complex exhibited the most favorable binding free energy, driven by strong van der Waals interactions, electrostatic complementarity, and stable hydrogen bonding. These interactions contributed to a tightly packed protein/protein interface and low root-mean-square-deviation (RMSD) observed during the last 50 ns of the simulation [1,27]. Conversely, Col_D-ImD and Col_E8-Im8 complexes showed weaker binding energies, depicted by fewer stable hydrogen bonds and higher hydration dynamics [1,28]. This represents the importance of intermolecular forces and interface hydration in guiding binding stability. Dynamic structural behavior was observed in Colicin D and Colicin E8 complexes, especially during the early stages of simulation following positional restraint removal. RMSD and RMSF analyses highlighted significant deviations, mirroring relaxation and interface adjustment. These adjustments are fundamental for the biological roles of these systems. For Colicin D, flexibility at the N-termini facilitates efficient interaction with bacterial tRNA [1]. Colicin E8’s dynamic nature aligns with its function as a DNase, which requires adaptability to engage bacterial DNA while avoiding host toxicity [1,24].

### 3.1. Overestimation of Computational Binding Energies Compared to the Experiment

Several factors may result in computational overestimation of the binding free energies of Col-Im complexes compared to that obtained from experimental data [22,29,30,31]. First, sampling limitations may contribute to the divergence of computational versus experimental binding energies [32]. The finite timescale of MD simulations and the number of independent simulations performed may result in gaps in the full exploration of relevant conformational space. Also, entropic contributions are, of course, incorporated into any experimental result. However, theoretical estimation of entropic contributions for large biomolecule complexes is difficult and computationally intensive, leading to simplifications (e.g., quasi-harmonic analysis [26]) or the exclusion of entropic contributions from simulations entirely. Additionally, standard force fields may not accurately reflect experimental conditions, leading to discrepancies between experimental binding energies and computational results [33]. Furthermore, differences between simulation parameters and experimental conditions (pH, ionic strength, etc.) could also contribute to discrepancies [34]. For example, water molecules buried at the interface of the protein complex may significantly affect binding energy as they may form a network of hydrogen bonds that shield and reduce electrostatic interactions between ligand (Im) and substrate (colicin). In addition, water has a fixed dipole (1.85 Debye) that can also act to shield electrostatic interactions between protein complex partners. These subtle dipole effects and hydrogen bond networks may be difficult to estimate with MD simulations and binding free energy calculations such that they accurately reflect experimental conditions. For instance, Zhou has raised the issue of the ionic strength of the solution having a significant effect on the association rates (*k*_a_) of the colicin E3-Im3 complex [35,36].

### 3.2. RMSF Maxima

Of particular interest is the RMSF maxima at residue 300 (**M**300) of Col_E3 and Col_E8 shown in the top panels of Appendix A, respectively. The maxima correspond to an unstructured loop region (AHDP**M**AG and AHDP**M**SG, respectively) resulting in a 180° hairpin turn of the colicin backbone chain that connects two 100 Å long coiled-coil antiparallel α-helices of the R-domain as shown in Figure 2a and Figure 3a. It has been proposed that this unstructured, primarily hydrophobic loop region of the R-domain serves as the initial point of attachment of the Col_E3/Im3 complex to the hydrophobic pocket of the BtuB outer membrane receptor of a bacterial cell [2]. A model was proposed that the loop region also acts as a molecular *hinge,* allowing subsequent separation of the two globular regions (rRNase-domain and T-domain) that surround and bind Im3 [30]. Once rRNase and T-domains separates and become in physical contact with the outer and/or inner membranes of the cell, the immunity protein, being less tightly bound, detaches having served its primary purpose.

### 3.3. Contribution of Interfacial Buried Water Molecules

Analysis of buried water molecules at the protein/protein interfacial region disclosed hydration dynamics as an essential factor in stabilizing the protein–protein interface. The complexes of Col_D and Col_E3 with their immunity protein were dependent on interfacial water molecules to maintain flexibility and adaptability, with hydration contributing to transient hydrogen bonds that stabilize the interaction. In contrast, the Col_E8 complex exhibited fewer interfacial water molecules, revealing a compact, tightly packed interface but potentially condensed flexibility [21]. The close matching of the number of buried water molecules in Col_E3-Im3 with previously published work [37] Also signifies its relevance and importance. These hydration approaches align closely with the complexes’ observed functional roles and differential stability. In addition, the consistent destabilizing effect of polar solvation energy in MM-PBSA calculations underlines a limitation of the Poisson-Boltzmann method, which overestimates the solvation contributions. Additionally, the cut-offs used for residue selection influence the per-residue decomposition, while the total energy accounts for the contributions from all residues. Improvements using tools like *DelPhi*, which provide grid-based solutions for electrostatic potentials, could help improve the solvation accuracy and, hence, binding energy predictions [12,21,38]. Expanding the studies with the available computational resources and integrating advanced sampling techniques with more accurate results help explore the conformational transitions more precisely [39,40,41,42,43]. Exploring these methods with some existing experimental validation will further elucidate the thermodynamic scenarios of these biologically relevant systems [21,24].

## 4. Materials and Methods

### 4.1. Bacterial Strains and Plasmids

The colicins and their cognate immunity genes were present in small plasmids carried by three different Shiga toxin-producing *E. coli* (STEC) strains that had been genomically sequenced previously [5,44]. The colicin E3 (PCG41550.1) and its cognate immunity protein (PCG41551.1) sequences were obtained from previous genomic sequencing of *E. coli* O113:H21 strain RM7788 [44]. The colicin E8 (AWJ52140.1) and its cognate immunity protein (AWJ52141.1) sequences were obtained from previous genomic sequencing of *E. coli* O43:H2 strain RM10042 [5]. The colicin D (AWJ36133.1) and its cognate immunity protein (AWJ36142.1) sequences were obtained from previous genomic sequencing of *E. coli* O103:H11 strain RM8385 [5]. Consistent with X-ray crystal structure data [2], eighty-three residues were removed from the N-terminus of both colicin E3 and E8. In addition, the N-terminal methionine was removed from the colicin E3 immunity protein. Sequences are provided in the Appendix A.

### 4.2. Structure Prediction with AlphaFold2

The structure of colicins and their respective cognate immunity proteins was predicted using AlphaFold2 (version 2.3.2) [16,45], which utilizes state-of-the-art deep learning techniques to predict protein structures accurately. The predictions were conducted on high-performance GPU resources provided by the SciNet Ceres and Atlas clusters (https://scinet.usda.gov). The amino acid sequences of each protein complex were submitted in FASTA format and processed through the AlphaFold2 pipeline. The system routinely generates multiple sequence alignments (MSAs), which are used as input for structural modeling. Out of the predicted models, the structure with the highest pLDDT confidence score and performance metrics was selected for each colicin-immunity complex. This high-confidence structure was used for molecular dynamics simulations and additional computational analysis. The results from AlphaFold2 were essential in further investigations into these protein complexes’ dynamics and interaction mechanisms. Additionally, the predicted structures were compared to experimental data to confirm their significance and precision for downstream applications [2,46].

### 4.3. Molecular Dynamics Simulations with GROMACS

Molecular dynamics (MD) simulations of the colicin-immunity protein complexes were conducted using GROMACS [28] (version 2024.1). The protein structures were obtained from AlphaFold2 predictions and prepared for simulations by adding hydrogens, solvating the system in a cubic box using the TIP3P water model, and neutralizing the system with appropriate counterions. All protonation states of polar residues were assigned based on the default CHARMM36 force field parameters at physiological pH (~7.0) without manual modification. This was done according to the standard protonation conditions and was consistent throughout the simulation. The CHARMM36 [47] force field was employed for the protein complexes.

Energy minimization was performed using the steepest descent algorithm until a convergence criterion of a maximum force of less than 1000 kJ/mol/nm was met. Following minimization, the system was equilibrated under constant volume (NVT) and constant pressure (NPT) conditions for 1 ns each. Temperature coupling at 300 °K was ensured using the V-rescale thermostat [48], while pressure coupling at 1 bar was applied using a Berendsen barostat [49]. Particle Mesh Ewald (PME) was employed to calculate long-range electrostatic interactions, and a 1.2 nm cutoff was applied for van der Waals and short-range electrostatic interactions. Subsequently, equilibration and production MD simulations were conducted for 150 ns using a 2-fs time step under periodic boundary conditions. Positional restraint was applied to Col_D-ImD and Col_E8-Im8 during the first 100 ns of the simulation to stabilize and prevent unrealistic structural deviation and facilitate system equilibration. System configurations were saved every 10 ps for further analysis. The stability of the protein complexes was confirmed by analyzing RMSD, RMSF, SASA, hydrogen bonds and radius of gyration. Interfacial residues were extracted from gmx_MMPBSA per-residue decomposition, and polar residues were counted. The equilibrated trajectory’s final 50 ns (100–150 ns of the entire trajectory) was used for binding free energy calculations. The number of buried water molecules at the protein–protein interface was also calculated within the trajectory with a cutoff distance of 0.25 nm, ensuring only tightly bound water within interface residues was included [50].

### 4.4. Binding Free Energy Calculations Using gmx_MMPBSA

The binding free energy of the colicin/immunity protein complexes was estimated through MM-PBSA approach using gmx_MMPBSA [51]. The Poisson-Boltzmann approach was employed to calculate the polar solvation-free energy, while the non-polar solvation-free energy was determined using the solvent-accessible surface area (SASA) approach. The analysis was based on 5000 snapshots extracted from the final 50 ns of the MD trajectories with frames saved every 10 ps. The binding free energy of each colicin with its respective immunity protein ($∆G_{bind}$) was computed using Equation (5).
(5)$$∆G_{bind}=∆G_{gas}+∆G_{solv}$$
where,
$$∆G_{gas}=∆E_{elec}+∆E_{vdw}$$
$$∆G_{solv}=∆E_{polar}+∆E_{non-polar}$$

Here, the total energy ($∆G_{bind}$) comprises the gas-phase molecular mechanics energy (Δ*G_gas_*) and the solvation-free energy (Δ*G_solv_*). The gas-phase energy consists of contribution from van der Waals (Δ*E_vdw_*) and electrostatic (Δ*E_elec_*). The solvation energy is separated into polar (Δ*E_polar_*) and non-polar (Δ*E_non-polar_*) components. Entropy contributions were neglected in this study to diminish computational cost and complexity, focusing merely on the calculation of the effective free energy components (gas-phase interactions and solvation energy) [22,51]. This approach was chosen to streamline the analysis and focus on the key energetic interaction driving binding. Furthermore, residue decomposition analysis was performed to identify the contribution of individual residues to the binding free energy, offering insights into the key residues involved in the protein–protein interactions. The gmx_MMPBSA.sh [51] script was used to compute the energy components.

## 5. Conclusions

We investigated the molecular determinants of colicin-immunity protein interactions, emphasizing the roles of binding energetics, structural dynamics, and hydration in stabilizing these complexes. Col_E3-Im3 complex is the most stable system, with binding free energy driven by van der Waals forces and electrostatic interactions. Also, Col_D-ImD and Col_E8-Im8 showed the initial dynamic behavior due to positional restraint removal, stabilizing over time with hydration dynamics playing a vital role. With the buried water molecules contributing to interaction flexibility and stability, dynamics hydration can also be identified as an essential factor in the colicin-immunity interface. Col_E3-Im3 and Col_E8-Im8 showed higher hydration, and Col_E8-Im8 demonstrated reduced hydration as it resembled its densely packed interface. This can be aligned with the functional roles of the complexes about their adaptability and specificity in bacterial competition. It can significantly impact food safety and microbial control because colicins represent natural antibacterial agents that can target foodborne pathogens. This study of the structural dynamics and binding behavior of colicin-immunity can lead the way for their role in biocontrol strategies, advancing agricultural and food production systems. Additional studies involving experimental validation and alternative computational approaches could further strengthen our understanding of these biological systems.

## Figures and Tables

**Figure 1 molecules-30-01277-f001:**
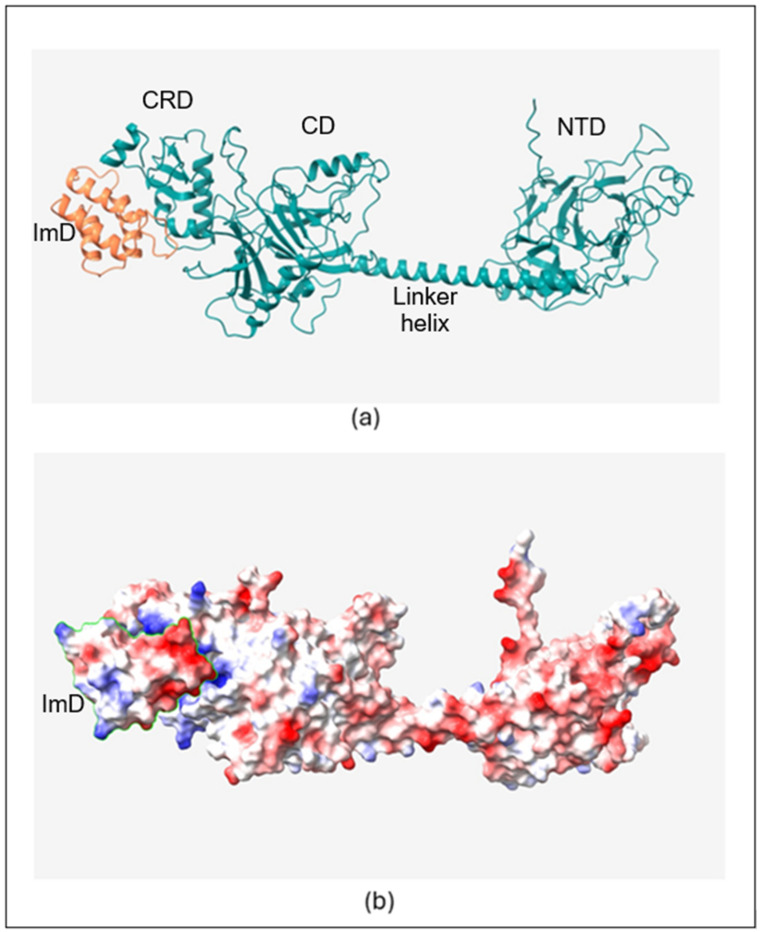
(**a**) AlphaFold2 predicted structure of Col_D-ImD. Central domain (CD), N-terminal domain (NTD), C-terminal RNase domain (CRD). (**b**) Electrostatic mapping on the complex with ImD outlined by the green line.

**Figure 2 molecules-30-01277-f002:**
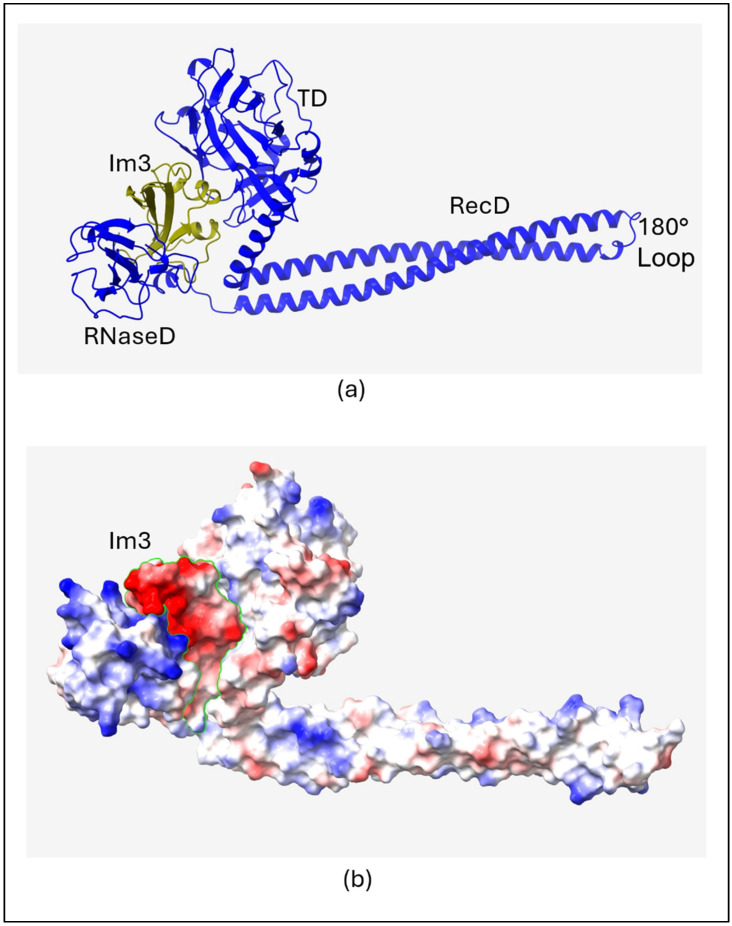
(**a**) AlphaFold2 predicted structure of Col_E3-Im3. Receptor domain (RecD), translocation domain (TD) and RNase domain (RNaseD). (**b**) Electrostatic mapping on the complex with Im3 outlined by the green line.

**Figure 3 molecules-30-01277-f003:**
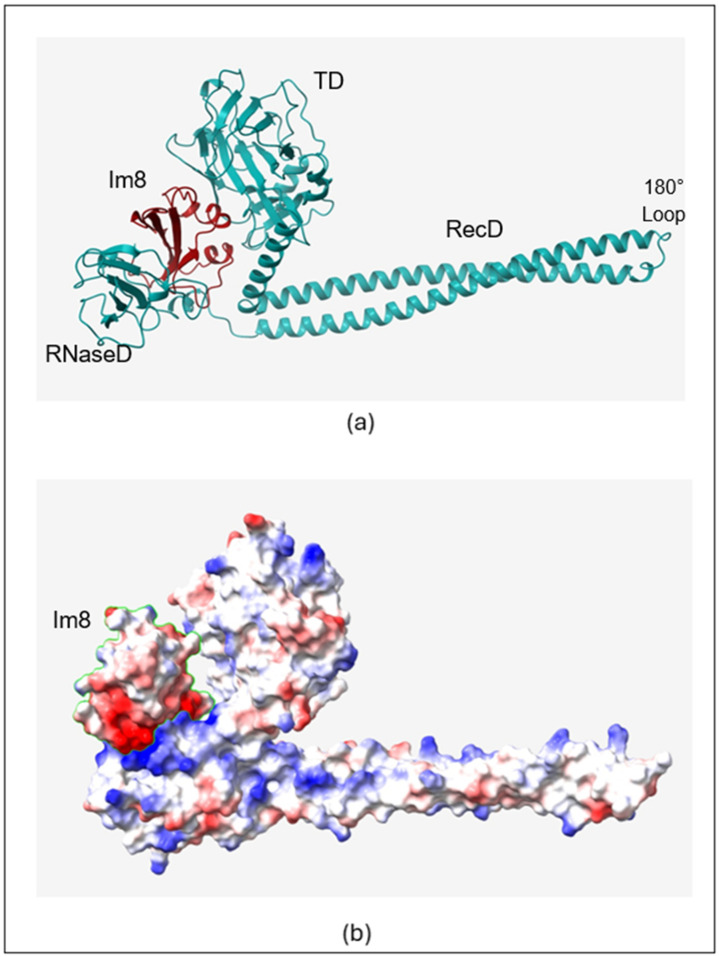
(**a**) AlphaFold2 predicted structure of Col_E8-Im8. Receptor domain (RecD), translocation domain (TD) and RNase domain (RNaseD). (**b**) Electrostatic mapping on the complex with Im8 outlined by the green line.

**Figure 4 molecules-30-01277-f004:**
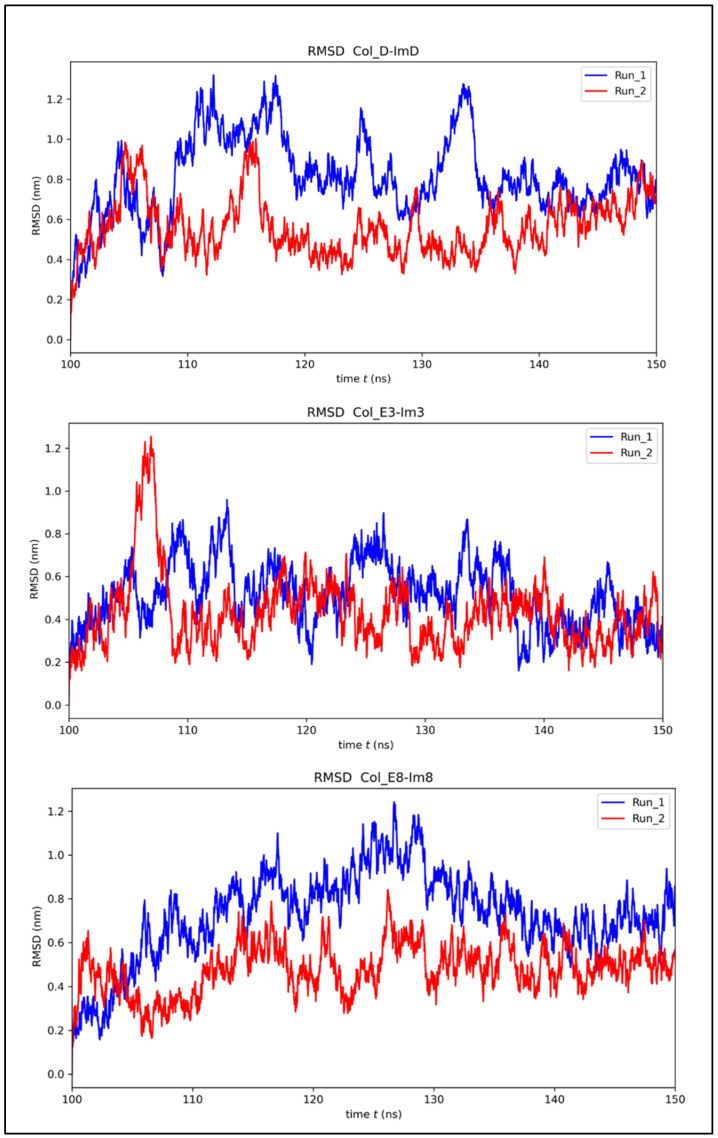
RMSD analysis of colicin/immunity complexes over 150 ns, showing stability in the 100–150 ns windows used for free energy analysis.

**Figure 5 molecules-30-01277-f005:**
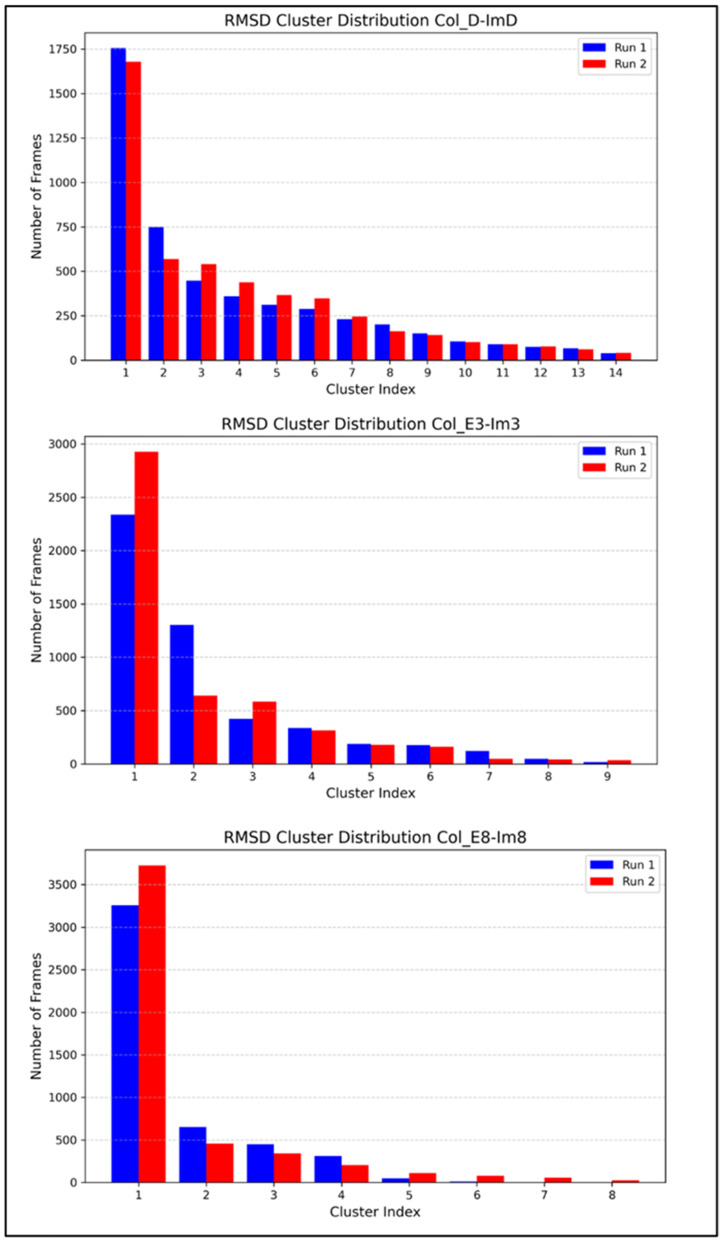
RMSD cluster analysis of colicin/immunity complexes in the 100–150 ns windows used for free energy analysis.

**Figure 6 molecules-30-01277-f006:**
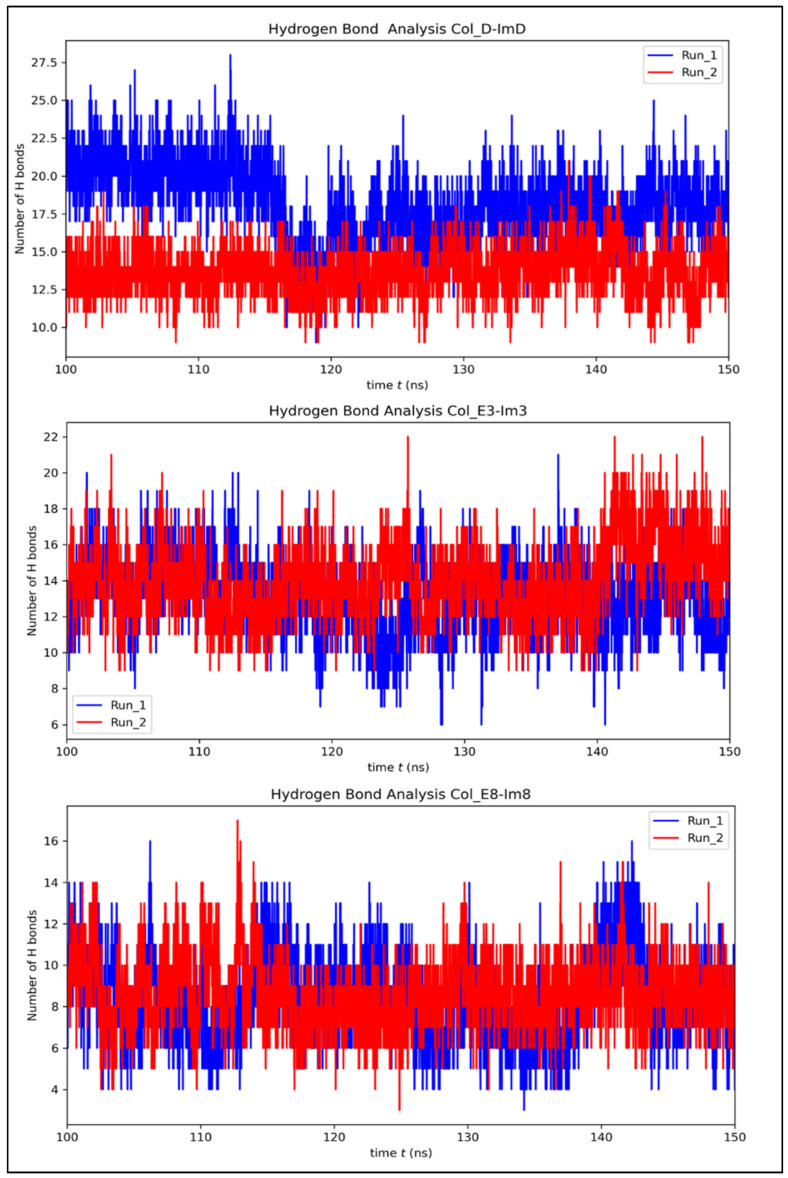
Hydrogen bond analysis of colicin/immunity complexes over 150 ns, showing stability in the 100–150 ns windows used for free energy analysis.

**Figure 7 molecules-30-01277-f007:**
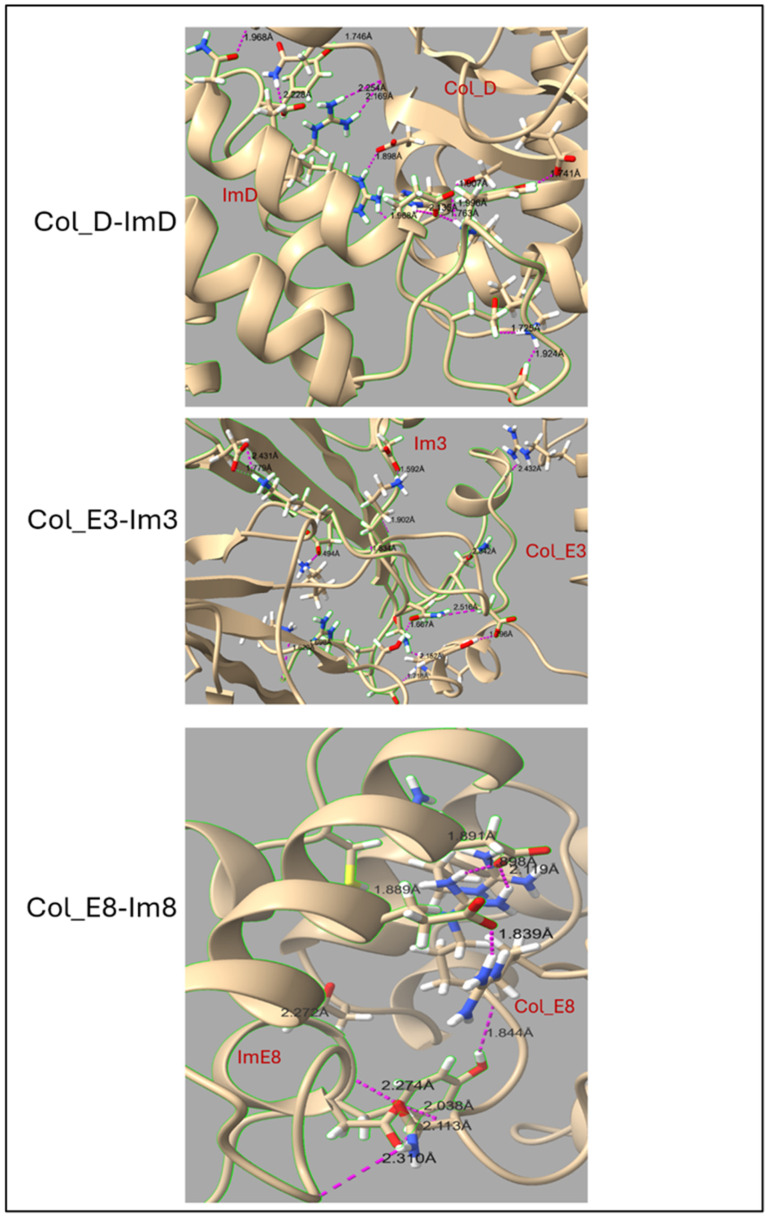
Hydrogen bonding interactions at the colicin/immunity interfaces. The hydrogen bonds are represented (dashed line) with their distance, and all immunity proteins are wrapped with a green selected line.

**Figure 8 molecules-30-01277-f008:**
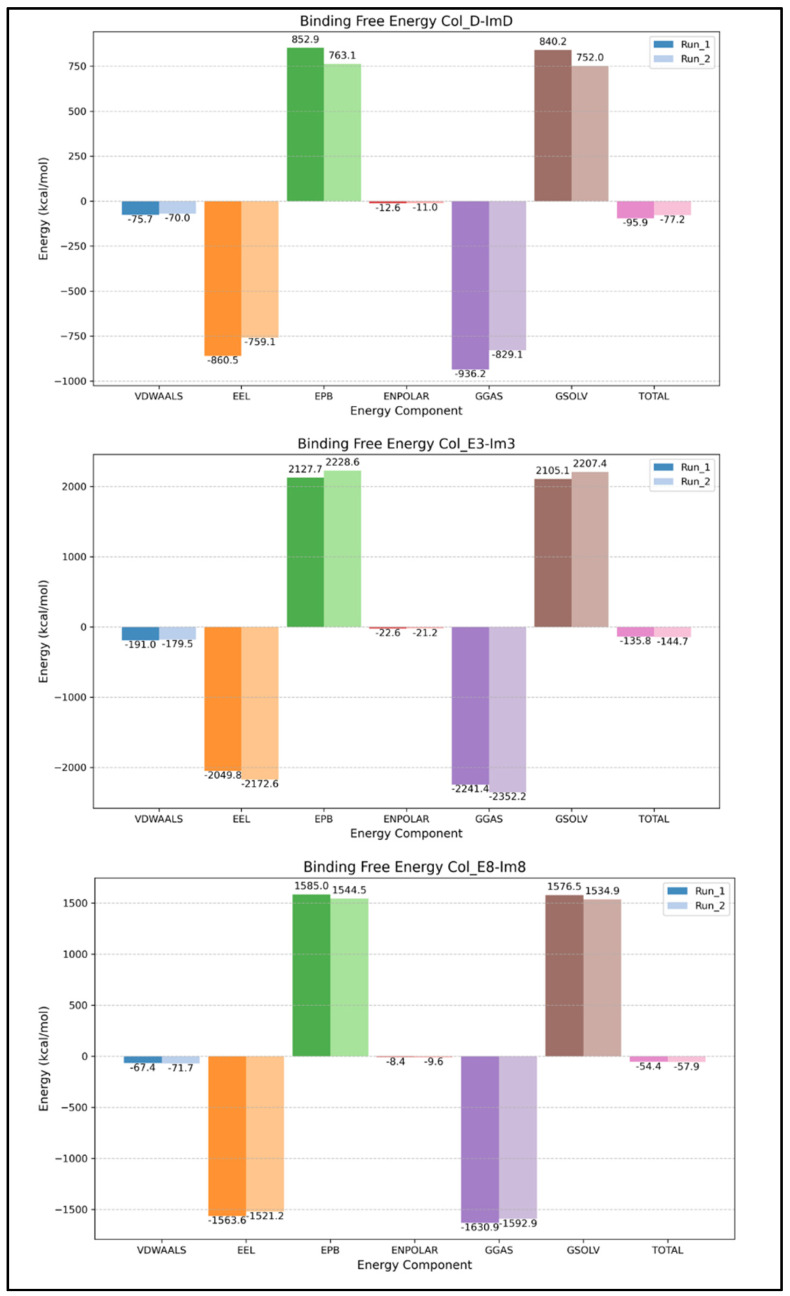
MM-PBSA results of colicin/immunity complexes.

**Table 1 molecules-30-01277-t001:** Intermolecular hydrogen bonds and buried (interfacial) water molecules.

Protein Complex	H-Bonds Before Simulation	Average H-Bonds After Simulation	Average Number of Buried Water Molecules
Col_D-ImD	13	15.83	20.15
Col_E3-Im3	11	13.56	23.55
Col_E8-Im8	10	8.50	11.81

## Data Availability

The data supporting this article have been included in the main text or as part of the Appendix A. The molecular dynamics simulation and binding free energy calculations files can be available on request.

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
