# Peer review of "Binding Free Energy Analysis of Colicin D, E3 and E8 to Their Respective Cognate Immunity Proteins Using Computational Simulations"

_molecules, 2025, doi:10.3390/molecules30061277_

Round 1

Reviewer 1 Report

Comments and Suggestions for Authors

In this work, the binding mechanisms of colicin-D, -E3 and -E8 with their respective immunity proteins (ImD, Im3 and Im8) are studied using AlphaFold2, MD and MM-PBSA approaches.

Results show that all three complexes exhibited the favorable binding free energy, with the Col_E3-Im3 being the most stable. Furthermore, electrostatics and hydrogen bonding enhanced the binding and stability, while polar solvation destabilized the binding. My major concerns are:

  1. How are the protonation states of polar residues determined for MD simulations?
  2. How many polar residues are there in the interface?
  3. Hydrogen bonds formed at the interface including those involving buried water molecules should be shown and their strengths should be compared.
  4. For MM-PBSA, what is the difference between two runs? The initial structures? Explain more.
  5. Some sentences need to be refined, e.g. the last sentence in 3.2.

After addressing the above issues adequately, the paper can be considered for publication.

Comments on the Quality of English Language

can be further improved. see the comments.

Reviewer 2 Report

Comments and Suggestions for Authors

The study presents a structural predictive analysis and the interactions between the antimicrobial proteins colicin D, E3, and E8 with their cognate host immunity proteins ImD, Im3, and Im8, utilizing computational tools such as the structural prediction tool AlphaFold and molecular dynamics for observing binding energy over time. The study addresses an interesting theme; however, it provides a rather superficial analysis of the proposed interactions. Longer molecular dynamics simulations were clearly warranted, as in Figure 4, the complexes did not appear to reach stability with only 150ns of simulation. Moreover, the authors seem presumptuous in claiming that binding free energy, van der Waals interactions, electrostatics, and hydrogen bonding are decisive in the protein complexes. It is important to recall that covalent bonds are more determinant in this context, and therefore, biological and chemical analyses are necessary to support a hypothesis. Overall, the article holds good potential, but the analyses, as presented, are superficial and contain methodological errors. I suggest that the authors, if intending to propose a purely computational study, employ additional analysis tools and establish controls for the analysis of molecular interactions.

Reviewer 3 Report

Comments and Suggestions for Authors

The length of reported all-atom MD simulations is too short for such systems to assess their properties. At least 500ns or even 1us production phase is required. Even coarse-grain simulations may be necessary, followed by all-atom simulations. The work may be considered after those changes.

Round 2

Reviewer 2 Report

Comments and Suggestions for Authors

The authors made significant modifications and clarified issues assertively in the text.

Reviewer 3 Report

Comments and Suggestions for Authors

The authors properly explained why the length of molecular dynamics simulations was appropriate for this level of study and pointed out the significance of the obtained results. Paragraphs regarding the additional trajectory analysis, added to the main text, support their approach. I believe the manuscript is in a publishable form now.